# Recycled (Bio)Plastics and (Bio)Plastic Composites: A Trade Opportunity in a Green Future

**DOI:** 10.3390/polym14102038

**Published:** 2022-05-16

**Authors:** Elisabetta Morici, Sabrina Carola Carroccio, Elena Bruno, Paola Scarfato, Giovanni Filippone, Nadka Tz. Dintcheva

**Affiliations:** 1Advanced Technologies Network (ATeN) Center, Università di Palermo, Viale delle Scienze Ed. 18, 90128 Palermo, Italy; 2Dipartimento di Ingegneria, Università di Palermo, Viale delle Scienze Ed. 6, 90128 Palermo, Italy; 3Consiglio Nazionale delle Ricerche, Institute of Polymers, Composites and Biomaterials (IPCB), Via P. Gaifami 18, 95126 Catania, Italy; sabrinacarola.carroccio@cnr.it; 4Consiglio Nazionale delle Ricerche, Istituto per la Microelettronica e Microsistemi (IMM), Via Santa Sofia 64, 95123 Catania, Italy; elena.bruno@dfa.unict.it; 5Dipartimento di Fisica e Astronomia “Ettore Majorana”, Università di Catania, 95123 Catania, Italy; 6Department of Industrial Engineering, University of Salerno, Via Giovanni Paolo II, 84084 Fisciano, Italy; pscarfato@unisa.it; 7Dipartimento di Ingegneria Chimica, dei Materiali e della Produzione Industriale, Università degli Studi di Napoli Federico II, 80125 Naples, Italy; gfilippo@unina.it

**Keywords:** recycling, thermoplastic, thermoset, composites, green economy, waste management

## Abstract

Today’s world is at the point where almost everyone realizes the usefulness of going green. Due to so-called global warming, there is an urgent need to find solutions to help the Earth and move towards a green future. Many worldwide events are focusing on the global technologies in plastics, bioplastic production, the recycling industry, and waste management where the goal is to turn plastic waste into a trade opportunity among the industrialists and manufacturers. The present work aims to review the recycling process via analyzing the recycling of thermoplastic, thermoset polymers, biopolymers, and their complex composite systems, such as fiber-reinforced polymers and nanocomposites. Moreover, it will be highlighted how the frame of the waste management, increasing the materials specificity, cleanliness, and a low level of collected material contamination will increase the potential recycling of plastics and bioplastics-based materials. At the same time, to have a real and approachable trade opportunity in recycling, it needs to implement an integrated single market for secondary raw materials.

## 1. Introduction

The excessive consumption of raw materials, the dwindling of fossil resources, the generation of plastic waste, and the uncontrolled plastic pollution are some of the major environmental problems that governments and academics must consider.

The development of biodegradable and compostable plastic products attracted the attention of the world and a lot of new and innovative products have already been put on the market [1,2]. However, one of the most significant ideas remains to improve the recycling that offers, simultaneously, the possibility of reducing the consumption of petroleum derivates that provide raw material and carbon dioxide emissions, thereby mitigating environmental pollution and simplifying organic waste collection, minimizing incineration and landfill issues [3].

Furthermore, due to a natural degradation in the environment, the plastic waste could become microplastics, i.e., very small-sized plastic, less than five millimeters long and ranging from 150 to 500 µm, thus, making it possible to be funded everywhere, and in particular, in sea water. As known, the microplastics are very harmful for environmental and organism health because they are already incorporated into the food chain from planktons to humans. The natural degradation of some synthetic polymers could take millennia, while for some last generation biopolymers, this process could happen in a short time, i.e., within a couple of months. To avoid uncontrolled micro- and nano-plastics formations, the urgency of correct and conscious plastic waste management and recovery has emerged. Although the natural degradation of biopolymers takes place within a short time, their biodegradation must be performed in controlled ways and locations, and before this, the bioplastic and synthetic plastic waste streams must be appropriately separated [1,2,3]. Therefore, the strategy to increase the supply of high-quality recycled plastic, i.e., to improve the efficiency in material production, needs to be considered one of the winning options.

Overall, plastics waste generated in the EU are subjected to landfilling and incineration at approximately 31% and 39%, respectively, and less than 30% is collected for recycling, according to Eurostat estimates (Report of EU Commission—A European Strategy for Plastics in a Circular Economy—(https://ec.europa.eu/environment/waste/plastic_waste.htm—Bruxelles, on 16 January 2018) [2]. In Figure 1, the plastic post-consumer waste rate of recycling, energy recovery, and landfill per country in 2018 are plotted. Moreover, the demand for recycling plastics, i.e., second-life materials, today, accounts for only approximately 6% of plastic demand in the EU because the profitable sector is low. 

However, according to the literature, a schematic diagram to illustrate and summarize the current status for the recycling of thermoplastics, thermoset, and their composites is shown in Figure 2a,b. It is evident that, for thermoplastic materials, all recycling solutions, i.e., from primary to quaternary recycling, could be carried out (Figure 2a) while, for thermoset and composites, the ternary and quaternary recycling are more appropriate recycling methods. For these materials, through ternary recycling, the monomers, oligomers, second-life materials, and reinforcement particles could be recovered efficiently and used to produce new manufacts (Figure 2b).

This review focuses on research studies in the last ten years regarding the primary, secondary, ternary, and quaternary recycling of thermoplastic, thermoset, and their micro-/nano-composites. Therefore, the first- and second-life materials are discussed, considering some target applications of these materials, and all discussed recycling approaches are referred to as a trade opportunity in a green future. Moreover, advantages and disadvantageous of different recycling approaches are addressed and future outlooks and critical aspects on which research would be required to further the full recycling of thermoplastic, thermoset, and their micro-/nano-composites. In addition, this review could be considered a valuable source, not only for academics, but also for the training of specialists in plastic waste management and for education of large audience readers.

## 2. Thermoplastic Polymers

Thermoplastics, such as polyolefins, polystyrenes, and polyesters, are versatile polymers made from linear molecular chains that soften on heating and harden on cooling to be easily molded in a wide range of manufacturing at high temperatures. Moreover, they are relatively inexpensive and weight-saving because the thermoplastics market has increased significantly over the last 70 years. Today, thermoplastic polymers occupy approximately 80% of the world’s plastic markets, and a large part is used to make disposable packaging items [4]. The life duration of the packaging or of other short-lived products can be so small that they are discarded within a year of manufacture; the drawbacks of the long thermoplastic polymers’ durability lie in their disposal and accumulation in landfills, creating severe environmental problems [5]. The advantage is that thermoplastics can be easily recycled and, if correct recycling is followed, they do not show drastic chemical property changes when heated or cooled multiple times.

The recycling of waste polymers is carried out through mechanical, chemical, and thermal methods (Figure 3). 

The recycling processes can be summarized according to the following four approaches:-the primary mechanical recycling or re-extrusion, usually set up next to the production line, which is a simple ‘in-plant’ recycling process of clean, uncontaminated single-type waste polymer usually considered off-specification: this represents a saving cost action to reuse otherwise lost material [6];-the secondary mechanical recycling is where the polymer is separated from other waste and classificated (sorting), the size usually reduced by cutting, grinding, or shredding, and then it is reprocessed (reprocessing), usually by melt extrusion. The drawbacks of this process are related to: (i) the need to separate the different types of plastics because of the poor compatibility between them if blended together; (ii) to the degradation process occurring during the process leading to an increase in coloration and poorest properties. In this latter case, the molecular weight of the recycled polymer is lower due to chain scission reactions caused by the presence of water, acidic impurities, or other different polymer-type presence, so the mechanical properties worsen; moreover, the products of thermal degradation are responsible for coloring. Strategies to avoid the polymer molecular weight drop during the reprocessing include intensive drying, reprocessing with degassing vacuum, the use of processing additives, such as stabilizers, chain extenders, etc. [7];-the chemical recycling, which is the tertiary or feedstock recycling, is the process which leads to the total depolymerization to the monomers, or partial depolymerization to oligomers, generally using temperature, pressure, solvents, and reagents, so that a new polymerization will take place to regenerate the original polymer [8]. The advantage is the opportunity to use the contaminated or mixed types of plastics and produce plastic with a higher quality with respect to what turns out from secondary mechanical recycling. The achievable goal is a high product yield in the depolymeritazion process and a minimal waste;-the quaternary recycling is related only to the energy recovery through incineration since the polymeric component is lost by thermal degradation. This method, however, may be ecologically unacceptable and hazardous for human health [9,10].

### 2.1. The Mechanical Recycling

The primary recycling of plastics, involving the use of one type and uncontaminated material by reconverting thermoplastic waste into its original pellet, enables having a recycled product with chemical and physical characteristics similar to those of the original one. This closed-loop recycling is practical when the polymer constituent can be effectively separated from contaminants and stabilized against degradation during reprocessing. Therefore, whereas this process is directly suitable for industrial wastes, secondary recycling using mostly post-consumer wastes, is not suitable for direct reprocessing due to contamination, which should be avoided to ensure not having products with inferior properties. The impurities catalyze the thermal degradation, i.e., cleavage reactions, while blending different plastic-types leads to phase separation during the processing because of the chemical incompatibility. These drawbacks mean poor properties of the recycled material, making it necessary to employ antioxidants, stabilizers, compatibilizers, coupling agents, and performance modifying techniques [11,12].

After collection and before reprocessing, a step of identification and separation of polymers is necessary to have better properties of recycled polymers. Manual sorting is very labor-intensive and high-cost, although it requires poor equipment and relies upon high accuracy. An alternative is to use mechanized sorting technologies. Different plastic polymer separation techniques are available based on different principles, such as gravity separation, flotation, electrostatic separation, magnetic separation, and sensor-based sorting [13,14].

The separation of plastics utilizing gravity or differences of densities is the most cost-effective process in automated industries. The separation of fractions having different densities is performed in a medium or slurry of intermediate density so that the light product floats and the heavy one sinks. The polyolefin fractions, such as polyethylene (PE) and polypropylene (PP), are light plastics having densities below 1 g/cm^3^ and they can be separated easily by density sorting using a process called float-and-sink. The separation is carried out in a flotation tank using water as the separation medium. Polyvinyl Chloride (PVC), Polyethylene Terephthalate (PET), polycarbonate (PC), and acrylonitrile–butadienestyrene copolymer (ABS), as heavy plastics, are considered unmanageable plastic mixtures because of their similar density. Equal density plastics separation is not possible by simple gravity methods, but it has to be carried out by flotation or triboelectric separation. In general, factors affecting separation by densities include: density difference with the medium; medium viscosity; particle form, sizes and surface texture; adherence of particles, agglomeration of droplets or bubbles with particles; hydroscopic resistance of plastics to sinking when at an air–water interface [15].

The froth flotation is a physical-chemical method, based on hydrophobicity differences between particles, successfully applied for the polymer separation. Because of the hydrophobic nature of most plastics, the addition of chemicals is required to change their critical surface tension to have an efficient flotation separation. Pita et al. investigated granulated post-consumer plastic, i.e., Polystyrene (PS), Polymethyl methacrylate (PMMA), Polyethylene Terephthalate (PET), and Polyvinyl Chloride (PVC) in the presence of tannic acid as the wetting agent, and the performance of the flotation separation of some bi-component plastic mixtures. Moreover, the effect of the size and shape of the particles was also analyzed. Results showed that the floatability of the analyzed plastics increased with the decrease in the tannic acid concentration, size of regular-shaped plastic particles, density, and lamellar shape. The results show that froth floatation is a suitable method for separating plastics with a particle size greater than 2.0 mm [16]. In another work, the floatation of PET from the plastic mixture of PVC, polycarbonate (PC), and acrylonitrile–butadienestyrene copolymer (ABS) was efficiently achieved, modifying the hydrophilicity the PET surface by potassium hydroxide (KOH) and ethylene glycol ((CH_2_OH)_2_) with the aid of sonication [17]. Significant efforts have been made by researchers to make the process efficient and economically attractive; furthermore, the rate at which plastic must be separated for the process has to be economically viable [18,19,20].

The electrostatic separation is a typical identification process used in mixed waste plastic separation and based on a surface charge transfer phenomenon. It is a typically two-step method: in the first, the mixed granular insulating materials are charged by triboelectric effect in a solid single-phase (rotating or vibratory, etc.) devices or gas–solid two-phase (fluidized bed, propeller-type, etc.) apparatus, for example, in a tryblo-cyclone, centrifugal forces are used to charge particles since they are accelerated and frictioned against internal walls [21,22]. Then, the second step involves the charged particles being subjected to an electrostatic deflection process, and then sorted in an electrical field thanks to Coulomb forces. There are different types of electrostatic separators, such as free fall, roll-type, and so on, where the charged granules are driven to opposite electrodes and then collected [23,24]. This technique is suitable for all types of plastic, although its limitation is related to the separable particle size.

Magnetic density separation (MDS) is a physical method based on differences in material density and it consists of four steps: wetting, feeding, separating, and collecting. The wetting and feeding step are performed to: (i) increase the hydrophilicity of the plastic surface; (ii) eliminate air bubbles; (iii) avoid air, which could cause turbulence in the fluid. The separation is obtained using a fluid magnetized by magnets located at the top and bottom of a flow channel, therefore, its hydrostatic pressure change, with the height position, led to a gradient of apparent density in the fluid; then, when the plastic particles mixture were introduced, the particles move to a zone where their mass density equals the apparent density of the fluid. The MDS allows to separate a multiple mixture continuously into the different components with high purity, in a single step and with the same fluid, so it is a very cheap and efficient separation method [25].

Laser-induced breakdown spectroscopy (LIBS) is an analytical technique based on a highly energetic laser pulse as the excitation source. In recent years, this technique has been more used than the well-proven near-infrared (NIR) technology because the latter cannot identify dark and black plastics, and in this sense, LIBS can be considered an evolution. The laser used in LIBS can form a short-lived plasma in the process. The excited species decaying to their ground levels emit electromagnetic radiations, that in turn, are detected. A complex spectrum, carrying information about the sample’s composition, is obtained. The detection time, laser energy, or measurement atmosphere affect the spectral appearance [26]. The drawbacks of the technique are related to the similitude of the polymer spectra because of the similar elemental composition. However, variations of the signal intensities due to the different stoichiometric ratios of the polymeric compounds can be detected by chemometric tools. Concerning the latter, it can also overcome the difficulty in identifying the presence of polymer additives or pigments [27].

X-ray florescence (XRF) is a non-destructive technique that uses x-ray produced by x-ray tubes, synchrotron, or radioactive material to irradiate a sample that then generates a fluorescent x-ray radiation with different energies levels corresponding to different colors. The technique is used for the determination of the chemical composition in different plastics materials and to detect elements such as Si, S, Cl, Fr, Ca, Ti, V, Fe, Cu, Zn, As, Kr, Zr, Nb, Mo, and Nb [28].

HyperSpectral Imaging (HSI) Technology is an innovative sensing method utilized to analyze solid systems in terms of composition and spatial distribution. The technique consists of the capturing and processing images at a vast number of wavelengths in VIS (400–700 nm) and VIS–NIR (400–1000 nm and 1000–1700 nm) wavelength ranges; in this way, it is possible to draw information about particle composition, morphological information, spatial and temporal fluctuations of the particle stream, etc. Spatial and spectral information, obtained simultaneously, are contained in a hypercube that is a 3D dataset with two spatial dimensions and one spectral dimension. Researchers demonstrated that HSI can be an effective tool for identifying unknown plastic in waste and the control quality of the polymeric feed, coming from other sorting procedures in the recycling process [29,30,31].

Many other techniques are used to identify polymers, but the process based on them could be a slow process and not applicable on an industrial scale for economic reasons.

The right choice of sorting technique in secondary mechanical recycling is needed to attain end products of suitable purity for reprocessing. The relative technologies also have to be manufacturing agreeable because multiple mechanical recycling could yield low-quality products and lead to a loss of recyclable materials. In other words, many factors of which can be attributed to the main techniques, including the composition of the plastic waste and the best and most economically available technology, have to take into account optimizing and making mechanical recycling economically attractive.

### 2.2. Chemical Recycling

Chemical (tertiary) recycling is the alternative to use for plastics unsuitable for mechanical recycling, such as film packaging consisting of several wafer-thin layers of different plastics whose clear-cut separation would be expensive, or colored plastic, such as common green or blue colored PET. Moreover, it is addressed as a valid option to mechanical recycling since it results in higher quality recycled material and higher tolerance to heterogeneous and contaminated plastic streams that could be treated with some pretreatments. The chemical recycling consists of a depolymerization to monomers (generation of the raw material) or pyrolysis to chemicals with different carbon numbers, useful for producing fuels or new polymeric materials. Biodegradable plastics can be composted, and this is a further example of tertiary recycling also described as organic or biological recycling [32]. Methods of chemical recycling include chemolysis or solvolysis, cracking, and gasification.

#### 2.2.1. Chemolysis

Chemolysis is a suitable process for treating homogeneous plastic waste. Similar to primary recycling, it is not directly appropriate for municipal plastic waste; in this view, to obtain a homogeneous stream, plastic waste has to be separated and sorted using various techniques as discussed above in the sorting section. Principal advantages are related to the final production of high value-added products and to the fact that it is an already operational process for PET [33] In chemolysis, depolymerization occurs with chemicals, and the most important procedures are glycolysis, aminolysis, methanolysis, alcoholysis and hydrolysis [34].

The glycolysis technique involves the use of the environmentally friendly solvent-reagent glycols and is the most used approach for the PET decomposition in which a transesterification reaction between ester group of the chain and a diol occurs in the catalyst presence [35]. Used glycols are usually ethylene glycol, diethylene glycol, propylene glycol, butylene glycol, and dipropylene glycol; the temperature range for the process is from 180–250 °C; moreover, the reaction needs to be catalyst-assisted [36]. The latter is a weakness of the process because used metal-based catalysts, such as zinc acetate, are hard to isolate so they often remain in the final products becoming harmful for human health and the environment. With the aim of overcoming the problem, researchers focus their research on finding more sustainable catalysts, such as ionic liquid [37] and urea-based deep eutectic solvents [38,39].

Aminolysis is a degradative process that uses various amines, such as methylamine, ethanolamine, butylamine, and ethylenediamine, in an operative range of low temperature (20–100 °C). This process is also proposed for PET degradation to produce diamides of terephthalic acid and ethylene glycol. Because of low temperatures, the kinetics are slow so a catalyst or microwave irradiation is required [40,41]. Among the catalysts tested, sodium acetate, deep eutectic solvent, and sunlight seem to be efficient and green at the same time [42,43].

Methanolysis implicates methanol use and relatively high temperatures (180–280 °C) and pressures. This process is largely used to degrade PET and PC. PET is merged with methanol and a catalyst, such as sodium methylate, and is then heated: this results in the reaction product dimethyl terephthalate (DMT) and ethylene glycol (EG). Methanol and EG can be rapidly recycled but the separation process and refining of the mixture composition from reaction products such as glycols, alcohols, and terephthalate derivates is very expensive [44,45]. The same issues occur in the methanolysis of PC to produce bisphenol A (BPA) and dymethil carbonate (DMC). A new low-energy catalytic process at ambient temperature was developed using potassium carbonate (K_2_CO_3_) as a catalyst for PET chemolysis [46], while ultrasound and ionic liquid was used for the greenest decomposition of PET and PC [47,48].

Alcoholysis and hydrolisis of PET have similar decomposition mechanisms and, while hydrolysis involves the use of water under different environments (alkaline, acidic, neutral), alcholysis employs methanol and ethanol, and is a process free of CO_2_. Recent research shows how microwave irradiation and ionic liquids also improve the efficiency of hydrolyses and alcoholyses [49].

#### 2.2.2. Cracking

In the recycling field, cracking could be defined as a process in which chemical bonds of the macromolecular chain are broken and so polymers result in smaller hydrocarbon molecules. This can be achieved through a process involving a reaction with hydrogen, known as hydrocracking or a reaction in an inert atmosphere by thermal cracking (pyrolysis) or catalytic cracking. Moreover, thermal cracking could even be plasma or microwave-assisted to have plasma pyrolysis and microwave pyrolysis, respectively.

Conventional pyrolysis is a simple technology and flexible process suitable for both low density and high-density polyethylene (LDPE, HDPE), PP PS, PMMA, for mixed plastic waste and for plastic that is difficult to depolymerize, including multi-layered packaging. In thermal cracking, the reactions occur by heating under the absence of oxygen to produce liquid oil, carbonized char, and volatile non-condensable fractions with high calorific value gas. In particular, PE and PP pyrolysis proceed through a random scission mechanism to generate a mixture of linear olefins and a wide range of molecular weight paraffins so a large variety of hydrocarbon products are obtained [50]. An unzipping mechanism occurs in the PS and PMMA thermal cracking to yield a high yield of constituent monomers [51]. However, the proportion of each produced fraction and their composition depends on the feedstock mix and quality, more than the process conditions. Temperature, pressure, residence time, and heating rate are the most important parameters that affect the final product and can be easily changed to optimize the product yield according to requirements. Thus, high temperatures (500–800 °C) lead to gaseous or char products, and lower temperatures (300–500 °C) produce liquid oil [52]. In addition, several types of reactor designs, such as fluidized bed reactors, batch reactors, vacuum pyrolizers, and kiln reactors have been investigated in the past because they play a fundamental role in affecting the residence time and having to overcome problems related to the low thermal conductivity and high viscosity of molten polymers [53,54]. Conventional pyrolysis is already an operational full-scale process for plastic waste with the major advantage of not causing water contamination; the drawbacks of the process are associated with the presence of PVC in the feedstock because the formation of chlorinate compounds in the oils and with the high energy requirement of the process itself. Moreover, the products often need upgrading before use so this process is cost-effective if high volumes are processed; alternately, the energy requirements of the pyrolysis could be covered by the combustion of the gas fraction produced or by concentrated solar power [55].

Plasma pyrolysis is a relatively innovative technology for recycling high plastic waste to produce a valuable synthesis gas (syngas) that integrates the pyrolysis with the thermochemical properties of plasma. The process uses extremely high temperatures (1730–9730 °C) in an oxygen-free environment to completely decompose plastic into syngas, composed mainly of simple molecules: CO, H_2_, and a small number of higher hydrocarbons [56]. The process is very fast, and the full decomposition is achieved so the produced gas is suitable for energy recovery being appropriate for electricity generation in turbines or for hydrogen production. The main advantage is the absence of toxic gases in the syngas, this is because the recombination reactions of gaseous molecules are avoided by quenching after the pyrolysis of plastic waste in a plasma arc reactor; on the other hand, a large amount of energy is required for the process and suitable plasma technology is still under investigation; therefore, nowadays, the scale of operation is at the laboratory level. Anyway, according to a sustainable hydrogen-based economy, plasma pyrolysis has great potential for the large-scale production of low-carbon hydrogen as a substitute for traditional fossils [57].

In microwave-assisted pyrolysis, a mixture of plastic wastes is added with a microwave adsorbent dielectric material and cracked into ethylene, propylene, and other useful chemicals by using microwave-assisted high-temperature pyrolysis. This process was developed by researchers looking for fast and energy-efficient heating to minimize the high cost of the conventional pyrolysis method due to heat loss. Utilizing microwave radiation as the heat source, energy can penetrate the inner part of the material providing high temperatures, high heating rates, and high heat transfer to plastic. The process allows controlling time and the magnitude of heating selecting the dielectric strength of the material to affect microwave pyrolysis efficiency [58]. Plasma assisted pyrolysis and microwave-pyrolysis is still a laboratory scale chemical recycling because of the variability in heating efficiency for different adsorbents and the poor information on the process itself makes it difficult to use on an industrial scale to date [54].

Catalytic cracking is a variant of the pyrolysis process since a catalyst, such as zeolites and silica-alumina, is added in order to save energy, reducing the temperature of the process because of a lower activation energy (300–350 °C) and increasing the rate of the reaction. Furthermore, the yield of products with high value increases: this process, indeed, narrows the hydrocarbon distribution, and oil with similar properties to conventional fuel is obtained [59]. The goal is the right choice of the catalyst to optimize the product’s distribution and selectivity, thus, several experimental studies and the testing of different catalysts have been carried out to enhance contact between polymer and catalyst and to improve liquid hydrocarbon yield. The catalyst is usually a liquid-phase contact or vapor-phase contact: in the former, the catalyst is contacted with melted plastics and acts mainly on the oligomers; in the latter, the contact is with the hydrocarbon vapors coming from the thermally degraded polymer. However, in order to increase the contact area to have better heat and mass transfer and overcome issues related to high viscosity, low thermal conductivity, novel catalyst, and nano-catalyst designs have been investigated [60,61]. Another trouble of the process is sensitivity to the contamination of plastics: the presence of nitrogen and chloride can deactivate the catalyst as well as the presence of inorganic particles blocking pores [52]. That is because the process could require a pre-treatment or to be limited to pure polymers, such as polyolefin. The catalyst used for PE and PP cracking are mainly those used in the petrochemical refinery industry and some catalytic processes are at an industrial scale, although the research for a catalyst that exhibits high activity, stability, selectivity, durability, recycling potential, and cost-effectiveness for a massive industrial implementation is still ongoing [62].

Interestingly, an innovative solventless catalytic process has been proposed for depolymerization of PET by a carbon-supported single-site molybdenum-dioxo catalyst (C/MoO_2_) to terephthalic acid (PTA) and ethylene. It has been demonstrated that this process is effective for both commercial and waste PET, and the catalyst exhibits high stability, at the considered condition of 1 atm of H_2_, and recyclability. Therefore, the C/MoO_2_ is an attractive catalyst for the chemical recycling of polyesters since it is selective, thermally moisture- and air-stable, earth-abundant, non-toxic, and recyclable. Further studies of scope and mechanisms are in progress. In addition, it is documented that the reaction of PET delopymerization is unaffected by the presence of polypropylene [63].

A currently published review by Kosloski-Oh et al. pointed out all the catalytic methods for chemical recycling or upcycling of commercial polymers. Specifically, they accurately overview and discuss the use of metal salts, metal catalysts, ionic liquids, solid acids, solid bases, organic photoredox catalyst, etc., as efficient medium for mono-/oligomers recovery from mostly used commercial polymers [64].

Hydrocracking, also called hydrogenation, involves a reaction with hydrogen on a catalyst in a stirred batch autoclave at moderately high temperatures and pressures (typically 350–500 °C and 30–90 atm) resulting in high product quality. Additionally, in this process, the catalyst plays an important role: several catalyst performances have been analyzed including those of nanocatalyst and transition metals supported on acid solids (such as alumina, silica-alumina, zeolites, and sulfated zirconia). These catalysts have both cracking and hydrogenation activities and, although gasoline product range stream information have been reported, not much is known about metal and catalyst surface areas, or sensitivity to deactivation [65,66,67]. Beyond the poor information, obstacles for the commercial scale of this operation are above all the high costs of hydrogen and the poisoning effect, such as that which takes place after the hydrocracking of PVC.

#### 2.2.3. Gasification

The conventional gasification process, suitable for mixed plastic, involves a partial oxidation reaction of polymeric waste with a gasifying agent (e.g., steam, oxygen, and air) at high temperatures from approximately 700–1300 °C to produce synthesis gas or syngas. The syngas is then used for the production of chemicals and fuel for the fuel cells to generate electricity. The composition of the syngas depends on the variety of feeding and on the oxidant agent, but high tar content and char are acquired in produced oil, so a further upgrading is needed before use. In general, air gasification leads to syngas with low calorific value because of atmospheric nitrogen presence. In order to enhance the calorific value, the air could be oxygen-enriched. Pure oxygen gasification results in a higher heating value, yet currently, this process is not cost-effective because it needs an upstream air separation unit. Lastly, steam gasification produces a high value and high hydrogen concentration syngas [68,69]. The result coming out from PET steam gasification experiments, conducted in a lab-scale bubbling fluidized bed, showed that high concentrations of CO_2_ and biphenyl are the prominent characteristics of PET steam gasification, in contrast to other common thermoplastics; moreover, increasing steam and temperature up to 800 °C improved the yields of H_2_ [70]. An example of a commercial-scale facility that converts waste into biofuel and renewable chemicals is Enerkem’s technology: the plant uses steam gasification in a fluidized bed reactor. Nevertheless, a high level of waste separation and a large amount of energy are required for the process, so economic considerations have to be accounted for [71].

Progress in the gasification process was achieved by plasma use: the products result in high purity and lower levels of tars. In the process, a plasma torch powered by an electric arc is used to ionize the gas, the untreated waste comes in contact with electrically generated plasma in the reactor and so the organic matter is converted into syngas and inorganic matter is turned into the by-product slag. The successful commercialization of plasma gasification can be achieved chiefly by reducing costs and increasing investigations [72].

It is important to underline that chemical recycling, often, needs a lot of chemicals and/or energy and is not suitable for all polymer types, so this process could be thought of as uneconomical and dangerous for the environment; moreover, the potential of specific chemical recycling technologies should be evaluated case-by-case and is not properly correct to generalize advantages/disadvantages of the technology in the chemical recycling approach. The right combination of chemicals and technical processes will determine the economic feasibility and the applications for the recovered material.

### 2.3. The Energy Recovery

Quaternary recycling is the most used method for plastic recycling today, being energy recovered from waste plastic by incineration. Municipal solid waste incinerators are a proven and robust technology familiar with mixed waste types of different origins. Incinerators meet with strong social opposition due to public health risks; on the contrary, it can help to reduce landfilling requirements and recover energy present in the materials being burned. The principal aim is a low organic emission rate from the furnace of the incinerator, but several studies are also concerned with the optimization of the technologies and of the process, parameters taking into account that a number of precautions have been pleased with particular regard in the plan whereas received waste, i.e., without pretreatment operation, are burned [73,74].

To summarize, in Table 1, all discussed recycling methods, which are applicable for thermoplastic materials, their short description, advantages/disadvantages, and status of the technology related to the current literature, are summarized; see below. 

## 3. Thermoset Polymers

Thermoset plastic is characterized by a three-dimensional crosslinked structure that gives them their useful properties that are heat and chemical resistant and highly resistant to loading. These features come with being difficult to recycle under simple heating and cooling cycles since they cannot be re-molded, in contrast to thermoplastics [75]. Currently, most thermoset polymers are landfilled or incinerated after their useful lifetimes, and thermosets, such as resins, polyurethane foams, and synthesized rubber, are recycled through chemical recycling and glycolysis hydrolysis; pyrolysis and microwave and ultrasonic treatments are some conventionally used methods. These methods typically require a high energy input, so many researchers focused on the design of reprocessable thermosets by using dynamic covalent bonds or cleavable crosslinks or cleavable monomers to achieve controlled thermoset degradation with a lower energy consumption. The dynamic covalent linkage shows the molecular mechanisms both associative (transesterification) and dissociative (Diels–Alder reaction) linkage, while the dynamic response of these bonds or monomers within their molecular structure is triggered by thermal, chemical, or optical external stimulus [76,77,78,79].

Tsuji et al. studied [80] the low temperature cleavage of double bond C=C through oxidative ozonolysis in an organic solvent; however, some researchers reported ozonolysis in aqueous medium [81]. In the latter, the process is more environmentally friendly and could be a valid alternative on a large scale.

Examples of cleavability by trigger are also reported for bio-based thermosets. They are mostly soft material as hydrogels and elastomer: it is not so easy to combine the high performance of thermosets with biodegradation.

To date, most of these new generations of degradable thermosets are negligible in an industrial approach because a complex expensive chemistry or poor mechanical properties of pyrolysis is considered to be one of the most optimal industrially available recycling treatments at present. The process has to be improved and that is because many research efforts are mainly focused on the reduction of pollutant emission, on the yield increasing, and on the energy consumption lowering, also analyzing the possibility to use a catalyst [82,83].

Solvolysis is less energy-intensive than pyrolysis and it is addressed as a successfully process for a wide range of resins, including solvolysis in a supercritical fluid [84,85] that is a good solvent because the high mass transport coefficients together with low viscosities; moreover, adjusting the applied pressure, the reaction rates and selectivity can be easily controlled.

Polyurethane foam can be recycled by way of the mechanical approach: the re-bounding, i.e., subjecting to heat and pressure a mix of powdered foam with binder, compression, and injection molding using shredded foam under high temperature (180 °C) and pressure (350 bar) or using carbamate exchange catalysis and twin-screw extrusion [86,87].

Vulcanized rubber is recycled with a devulcanization process that is the cleavage of the S-C bond by thermomechanical recycling, recycling under irradiation (microwave or ultrasound), or biological recycling. The thermomechanical method is conducted with high shear and high temperatures in a high shear cone or in a twin-screw extruder [88]. In biological recycling, some aerobic or anaerobic microorganisms decompose rubber feeding of sulfur. This process is very simple, up-scaling, economic, and eco-friendly, but it needs a detoxification process: compounds harmful for microorganisms have to be removed before the biological recycle [89].

Additionally, for thermosetting recycling, every technology has its own advantages and disadvantages, and it is crucial to associate the right recycling process with the application of the recycled product, see Table 2.

Summarizing, the thermoset polymers can be recycled by chemical processes that are able to destroy the three-dimensional crosslinked structure acting through chemicals and/or suppling energy to networks, see Table 2. As cited, thermoset materials are currently subjected to landfilling and incineration.

## 4. Thermoplastic and Thermoset Micro-/Nano-Composites

Micro-/nano-reinforced polymer-based composites are a class of composite materials in which the reinforced phase is dispersed or structurally integrated in a continuous polymeric phase [90,91,92]. The host polymer matrix has two important roles, proving geometrical stability for composites and preserving the reinforcement agents from environment, isolating gas and humidity. The dispersed phase has a reinforcement role and consists of particles or fibers, having different aspect ratios, i.e., shape, dimensions, length-to-diameter, etc., and chemical nature, i.e., inorganic or organic nature, surface modification and functionalization, etc. Reinforcement particles and fibers could be randomly dispersed or oriented into the host matrix, providing greater mechanical performance for composite materials. Furthermore, to understand the effect of added particles and fibers on the matrix properties and performance in-depth, the interactions and/or reactions between the particles and host matrix must be investigated and established [92].

Therefore, the formulation of micro-/nano-composites using thermoplastic and thermoset matrices gives the opportunity to produce materials that are suitable for engineering applications with significantly improved properties, such as mechanical resistance, i.e., elastic and compression moduli, impact strength, etc., resistance to abrasion, barrier properties towards gas and humidity, and in some cases, improved durability [90,91,92,93,94]. 

However, to achieve tailored engineering applications, the thermoplastic polymers, such as high-density polyethylene (HDPE), polypropylene (PP), polyamides (PA), acrylonitrile butadiene styrene (ABS), polybutylene terephthalate (PBT), and polylactide (PLA), can be successfully reinforced using particles and fibers, while thermoset matrices mostly considered for engineering composite formulations are based on cured epoxy resins [93].

The frequently used particles as reinforced agents for both thermoplastic and thermoset matrices are particles, having different shapes and aspect ratios (i.e., spheres, tubes, roads, etc.), and dimensions (i.e., micro or nano sizes), such as calcium carbonate, mica, silica, aluminosilicates, layered double hydroxides, metal oxides, etc. [90,91]. Moreover, currently published studies propose the preparation of novel char particles, which are mainly composed by carbon atoms at ca. 90%, by slow pyrolyzes processes of biomass and/or agricultural waste, as efficient reinforcement agents for polymers [95,96]. 

Commonly considered fibers, useful for both thermoplastic and thermoset matrices, are glass fibers, carbon fibers, and naturally occurring fibers, such as wood, flax, cotton, bamboo, etc. As known, the glass fibers are cheaper and better performing reinforced fibers [97,98], while the carbon fibers are more expensive and provide excellent mechanical performance, especially when oriented [10]. According to the literature, the glass and carbon fibers can deliver to composite manufacts, with excellent performance, low weight, and good processability, making these materials suitable for several engineering applications in the different sectors, such as aerospace [99], automotive [100], energy [101], and the construction industry [102]. In the last two decades, the use of natural fibers for the formulation of polymer-based composites has attracted significant attention from both industrial and academic words and numerous studies are published investigating the performance and properties of these eco-friendly composites in-depth [103]. Although, natural fibers are very attractive, due to their eco-friendly nature, the introduction of fibers in some polymer matrices could induce a depletion of oxygen resistance of these composites due to the complex chemical composition of fibers and matrix crystallinity reduction [103].

As expected, the presence of micro-/nano-sized particles and fibers has certain beneficial effects on first-life polymer-based composite materials. Unfortunately, the presence of these additives could negatively influence the materials’ circularity. To resolve it, there is the need to design suitable recovery processes. If the unfilled polymers and filled ones are presented in the collected mixed waste stream, they are incompatible, although being of the same matrix materials. Therefore, the primary recycling of filled polymer materials can be considered a viable way, although the particles’ amount/content and polymer matrix degradation, undergone during the first production step/manufacturing, must be tuned and controlled to formulate useful second-life materials. The presence of filled polymer-based materials in the waste stream, containing different types of particles and amounts, makes the secondary recycling almost impracticable. In contrast, the ternary and quaternary recycling could be considered more appropriate methods for these systems (Figure 2b). In addition, the micro-/nano-fillers, if the composites are subjected to landfilling, could have a negative environmental impact because of their chemical inertia and impossibility to degrade, resulting in resource losses.

Further, successfully recycling micro-/nano-composite materials, the particles/fibers, and matrices must be separated and considered as resources for second-life materials. The separation processes can be carried out through matrix solubilizations, depolymerization reactions, or slow pyrolysis. In some cases, during the separation process, both particles/fibers and matrices could experience dimensions, compositions, and property alterations, which could compromise their use as efficient second-life reinforcement additives. 

### 4.1. Mechanical Recycling

However, the mechanical approaches are based on common techniques that can be encountered in the primary and secondary recycling of composite materials. For example, the entire composite materials were firstly ground to obtain the feed for the manufacturing and then processed by different techniques, such as compression molding, injection molding, 3D-printing, etc. As documented, Colucci et al. [104], and Pietroluongo et al. [105] employed injection molding to prepare the recycled composite specimens, Voltz et al. [106] recycled the composite by extruding multiple times on a twin-screw extruder before being injection-molded to prepare specimens, while Kiss et al. employed compression molding [107]. Therefore, a common drawback to the mechanical recycling of composites is the grinding steps that cause the shortening and damaging of reinforcement particles and fibers. Generally, all the reviewed works observed size and/or composition variations and shortening of the reinforcement particles and fibers after mechanical recycling [104,105], and for this reason, this approach is not fully applicable, especially, at the scale. Considering these inconveniences, according to the literature, the use of virgin polymers as matrices for second-life manufacts and/or the introduction of compatibilizers improving the adhesion between host matrices and reinforcement agents, have been currently proposed, but also these solutions have limited applicability at the large industrial scale [106,107]. 

Moreover, the mechanical recycling of thermoset composites is a tricky matter, and it is even harder than thermoplastic composites. Usually, currently used thermosets are epoxy resins, based on cured di-glycidyl ether bisphenol A (DGEBA), which is one of the most common commercial resins for particles and/or fiber particle-reinforced manufacturing [108,109,110,111,112,113]. The thermoset composites, when subjected to pretreatment operations, such as grinding, shridding, etc., for mechanical recycling, underwent significant particles/fibers-damaging because of the 3D-structure of matrices and high values of mechanical stress imposed to structure disassembling. Based on this, the academics are currently searching for alternative approaches that prevent the variations of sizes, dimensions, and compositions of reinforcement particles and fibers during recycling steps. 

### 4.2. Chemical Recycling

The chemical recycling of composites, through matrix solubilization and reinforcement particles and fibers recovery, is an alternative approach to mechanical recycling. Therefore, the use of solvents and their recovery through vacuum distillation or similar processes, as well as relatively high temperatures and pressures and long dissolution times, makes the chemical recycling scientifically valuable but at the same time an expensive and non-eco-friendly method for composites recycling, and even more if applied at a large industrial scale.

Interestingly, Okajima et al. used supercritical methanol at 285 °C and 8 MPa to break the ester bonds between the epoxy backbone and the crosslinking chains of the matrix [114]. Wang et al. [108] recycled an amine-cured di-glycidyl ether bisphenol A, containing carbon fibers, by selectively breaking the C-N bonds between the crosslinks and backbone. Rather than using supercritical fluids, the authors employed AlCl_3_ as a catalyst and acetic acid as a solvent at 180 °C. Another strategy includes selectively breaking the C-N bonds employed by Liu et al. [115] to recycle epoxy-based, carbon fiber-containing production scraps from the aerospace industry. Overall, the strategy to chemically recycle the thermoset composites provide networks to bear covalent bonds that can undergo bond exchange reactions, enabling the materials to be temporarily de-crosslinked, reshaped, and recycled.

### 4.3. Energy Recovery

Energy recovery is an alternative method for recycling, and it can be considered a valuable method because of energy demand growth. The drawbacks of this method are related to the high temperatures involved (up to 550 °C) that can damage the particles and/or fiber phase and the development of CO_2_ and other gas [116]. Therefore, the recent shift in policy towards sustainability and circularity in the European [117] and worldwide economy [118,119] suggests that the energy recovery of composites must be gradually replaced by materials recovery, minimizing the damaging of materials and the production of greenhouse gas. To sum up, the energy recovery of thermoplastic-based composites must be considered as a last solution/approach, after materials recovery, for the management of the polymer-based composite waste stream. 

An overview about the recycling strategies that could be considered for polymer-based composites are summarized in Table 3, below. As mentioned above, the main problem for the recycling of these materials is related to their collection together with unfilled polymer-based materials. For a correct reprocessing of polymer-based composites, they must be sorted and recycled separately from other polymers.

## 5. Biopolymers and Biopolymer-Based Micro-/Nano-Composites

In the last two decades, the formulation and production of biodegradable and/or bio-based polymers, as alternative to the synthetic counterparts, is a viable strategy towards sustainability in a green future. In addition, the formulation of biopolymer-based composites, using thermoplastic bio-based and/or biodegradable polymers and naturally occurring reinforcement particles and fibers, has attracted great attention for both academic and industrial worlds [1,2]. The most commonly used biodegradable thermoplastics, also for composites formulations, are aliphatic biopolyesters, specifically, polylactic acid (PLA) [120]. Therefore, despite the interest in the growth of biopolymers, their waste stream is limited, and as known, PLA and/or PLA-based composites are not collected separately from other polymer streams [121]. Obviously, this negatively influences the recycling status of biopolymers and the biopolymer-based composites, and there is much activity in patenting, rather than in publications; the information is not found anywhere else. 

However, as expected, the recycling process and technology for all biopolymer composites, being recently formulated composites, is addressed currently in patents and scientific papers, and not yet on a large industrial scale. Therefore, the main recycling process, examined here, includes sorting, mechanical recycling, chemical recycling through hydrolysis, alcoholysis, and thermal catalytic-assisted depolymerization, and recently proposed enzymatic depolymerization. 

Biodegradable polymers are susceptible to be broken down into simple compounds because of microbial action, and different bioplastics have been known to undergo this process in a reasonably short time (e.g., six months), and are commonly identified as biodegradable, though to substantiate biodegradability claims, certain standards have been put into place in the past twenty years; the main International Organization for Standardization, ISO, [122] and European Committee for Standardisation, CEN, [123] standards in place, many of which are shared as the CEN standards are often based on the ISO ones.

As known, the definition of biodegradation is exclusively focused on the biotic phenomena, and it is important to remember that abiotic phenomena take place during the biodegradation of a polymeric material, and these can have a strong influence on the overall degradation rate. Three different steps can be identified through which biodegradation occurs: (i) the first step is referred to as biodeterioration, where the material is broken down into smaller fractions due to biotic and abiotic activity; (ii) the depolymerization step by enzymes, during which the polymer chains are broken down into shorter oligomers and eventually monomers, though this process can also result from abiotic phenomena; (iii) the third step of biodegradation comprises of the assimilation and mineralization processes, during which monomers and oligomers from the broken-down polymer can reach the cytoplasm and enter the metabolism of the microorganisms [124,125]. 

Therefore, Cosate de Andrade et al. [126] presented a Life Cycle Analysis of PLA comparing chemical recycling, mechanical recycling, and composting, and they found that mechanical recycling had the least environmental impact, followed by chemical recycling and, lastly, composting, when considering the climate change, human toxicity, and fossil depletion categories.

Lamberti et al. [127] cover the recycling routes of several bio-based polymers, and overall, they observe that the best practice is to reuse any plastic as much as possible before recycling, and then apply two stages of recycling: first, mechanically recycled until the resulting material is of commercially acceptable grade, and second, chemically recycled to recover part of the original monomers. Finally, the authors conclude that the biopolymers’ mechanical performance needs to be improved and that better schemes for recycling and waste collection need to be put into place, see Table 4.

To sum up, the recycling of bio-based polymers and their composites must be further investigated and better addressed, considering the target applications for these materials, see Table 4. In addition, for a successful recycling process, the bioplastic and their composites waste stream must be collected separately to the other plastic waste stream. 

## 6. The Waste Management and the Market of Recycled (Bio)Plastics and (Bio)Plastic Composites

Today, solid waste management remains a concern for local administrations due to inadequate legislation and poor widespread correct information on waste disposal. The knowledge of an application of the 4R rule (i.e., Reduce, Reuse, Recycle, and Recover energy) is mandatory to minimize waste in the environment. However, to have a real benefit from recycling, the material disposal must be correctly carried out; for example, mixing recyclable and non-recyclable plastic in the same box would mean that they have to be sorted again if collectors do not otherwise the property and the value of the recycled plastic will be impacted.

The quality of the recycled polymers is indeed influenced by polymer cross-contamination, additives, non-polymer impurities, and degradation [128]. However, to summarize the currently applied strategies for recycling processes, i.e., primary, secondary, tertiary, and quaternary recycling of (bio)polymers and their composites (Figure 4) an overall schematic illustration is reported.

A combination of different technologies, a collaboration across the population and the institutions plus an integration of technical and economic involvement, is required to develop impactful solutions and successful solid waste management.

Another topic is related to the market of recycled plastic. According to the New Circular Economy Action Plan report approved by the Environment Committee of the European Parliament, an integrate, functionalized, and high-quality single market for secondary raw materials needs to be implemented so that an increase in plastics recycling rates will occur and the transition from a linear to a circular economy will turn more easily. For a real growth strategy for Europe, it is essential to implement existing EU legislation to avoid environmental costs associated with illegal waste export or landfill that bring into dangerous non-realized circular economy market developments and to costs which have been estimated by the European Commission at being up to EUR 4.8 billion annually. On the other hand, fiscal incentives aiming to support investment in infrastructure and demand for recycled material may be a signal to increase their attractiveness and improve their trade opportunity.

Finally, another problem in the recycling of post-consumer plastics is related to the high cost of transportation and collection since the plastic packages and containers occupy a large volume; therefore, the proposed solution calls for plastics manufacturers to assist localities in financing grinders and other equipment that reduce plastics into a denser form that is more economical to transport.

Recycling plastic waste is a real opportunity not only for waste management but also for developing a green, circular, and sustainable economy and for the development of new performing high-value polymeric materials from waste.

## 7. Conclusions

In the present review, we reported on the possibility of recovering polymer materials, oligomers, and/or monomers, and micro-/nano-particles from the recycling of (bio)plastics (thermoplastics and thermosets) and (bio)plastic composites. The number of papers is limited (one hundred twenty-eight in total) compared with the enormous, published research documents on polymers, biopolymers, and composites recycling, but this review would highlight a trade opportunity in a green future; it would also result in a valid source for education activities. Therefore, recycling could offer opportunities for the efficient recovery of materials and/or energy and the possibility to design improved second-life manufacts, reducing environmental impacts and offering more job opportunities.

Regarding the recycling of thermoplastic materials, all recycling strategies are applicable, e.g., from primary to quaternary recycling, and an appropriate design of second-life manufacts must be performed considering the decrease in properties of materials as a function of the reprocessing steps. As discussed, it is very important to consider the nature of collected materials, e.g., homogeneous or heterogeneous nature/composition, to be subjected to mechanical recycling, and obviously, this has vital importance on the sorting, cutting/grinding/shredding, and reprocessing operations. Therefore, also for chemical recycling, the nature of collected materials has vital importance. For example, only homogeneous (bio)polymers materials could successfully perform the chemolysis through glycolysis, aminolysis, methanolysis, alcoholysis, and hydrolysis. Conversely, for heterogeneous materials, the cracking and gasification could be considered more appropriate methodologies. Finally, the collected (bio)plastic materials, having an extremely heterogeneous nature, could be successfully recycled through energy recovery.

Regarding the recycling of thermoset materials, since they are usually 3D-structured and contain different chemical bonds, e.g., amides, esters, etc., and interchain bridges, their recycling process could be successfully designed by applying chemical and/or energy recovery approaches. Therefore, for the ternary recycling of thermoset materials to be considered a sustainable approach, there is a need to think about the use of solvents and mediums for chemical depolymerization and for monomers/oligomers recovery. As expected, this approach is limited in sustainability and efficient materials circularity.

Regarding the recycling of micro-/nano-(bio)plastic composites, most research is focused on the recovery of the fiber phase, with the polymeric phase being completely down-cycled to fuel. Taking into account the shift of worldwide polices towards sustainability and materials circularity, the closed loop approach for the recycling of composites will become increasingly important. As is well-known, the (bio)plastic composites have significant use in numerous daily life manufacts, automotive components, devices for energy recovery, etc., and their use continuously increases, encompassing in emergencies for their correct management and collection at end-of-life. Unfortunately, the (bio)plastic matrix and (bio)plastic composites, although being the same plastics, are incompatible because of the impossibility of controlling and tuning the reinforcement concentrations and degradation undergone by the matrix during reprocessing steps, and for this reason, the micro-/nano-(bio)plastic composites are particularly suitable for ternary and quaternary recycling. Finally, the resistance against oxidation of these complex systems plays a determinant role in terms of applications of second-life composites, and their durability must be considered for the design of appropriate materials circularity.

However, the recycling processes of polymers, biopolymers, and their composites could be considered real opportunities, if all the materials are appropriately separated and their degree of degradation/oxidation level undergone during the service conditions evaluated. Then, based on the specific characteristics, the separated polymer materials must be subjected to more appropriate recycling operations through primary, secondary, tertiary, and quaternary recycling.

Summarizing, the actual trend is for the amount of recycled material to increase because it is required to decrease the amount of plastic in landfill and in the environment, since it is potentially harmful for the world, and to recover raw material and energy. Therefore, the proposed policy regulations for waste management, as well as the governmental investment, are slowly going to promote and support plastic recycling. Another key factor seems favor the recycling direction, which is the market accepting this new competitive recycled product; the demand for recycled plastic items are going to increase because raw material depletion involves the increasing material price; moreover, consumer focus on environmental issues and sustainability are increasing.

The development of new products looking to this sustainable approach is strictly related to the development of new technologies able to make the process of recycling plastic faster, easier, industrially scalable, and economically attractive. The largest technological gaps include the high cost of collection and sorting of waste plastic, and as such, the high cost of some chemical environmentally friendly recycling technologies. Hence, there are opportunities and challenges for researchers and for industry in the recycling field, including:-providing more efficient collecting and sorting in order to separate additives and matrices and to recover both of them;-finding new green and more economic chemicals to use following chemical recycling approaches;-using low value or unused plastic, which would otherwise be lost and wasted;-producing higher quality products that can compete with virgin materials on price and quality;-recovering energy and monetizing the waste protecting and helping the world.

## Figures and Tables

**Figure 1 polymers-14-02038-f001:**
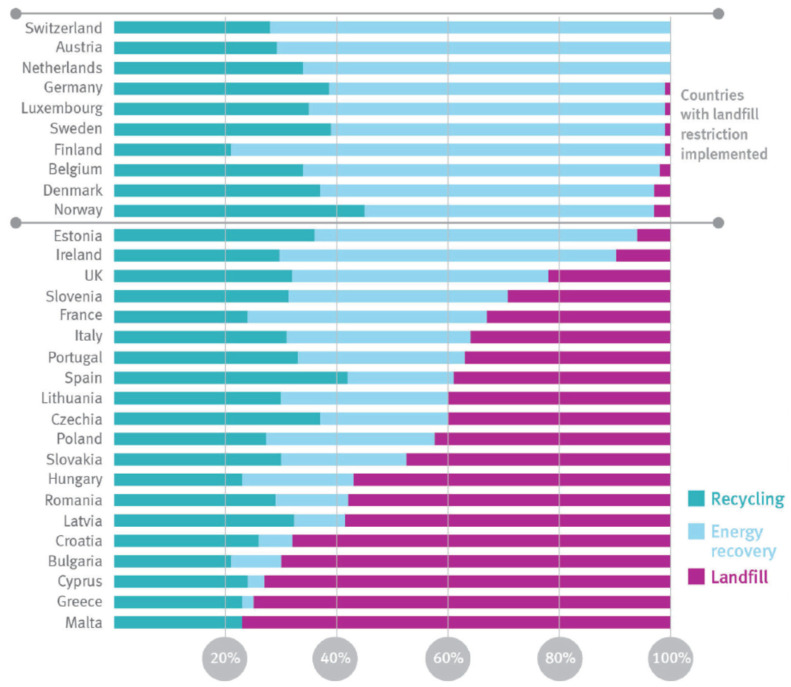
Plastic post-consumer waste rates of recycling, energy recovery, and landfill per country in 2018. Source: Plastics Europe. Plastics—the Facts 2019 [2].

**Figure 2 polymers-14-02038-f002:**
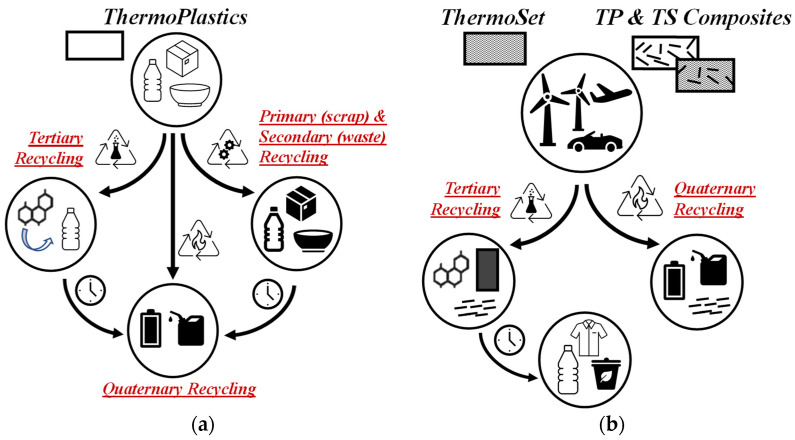
Schematic diagram to summarize current status for recycling of (**a**) thermoplastics and (**b**) thermoset and the composites, highlighting more appropriated and applicable at large industrial scale recycling strategies.

**Figure 3 polymers-14-02038-f003:**
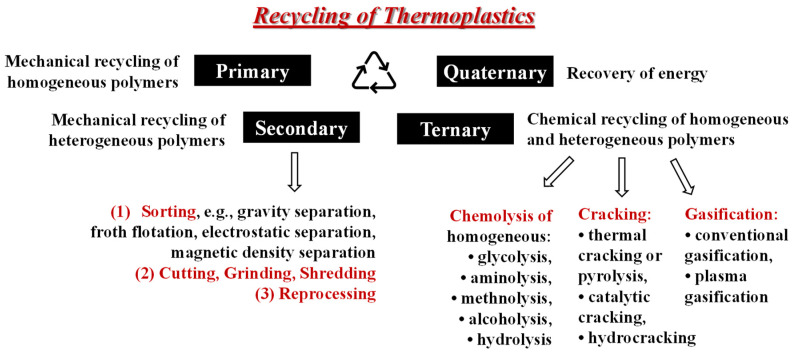
Schematic diagram to summarize the recycling processes for thermoplastics.

**Figure 4 polymers-14-02038-f004:**
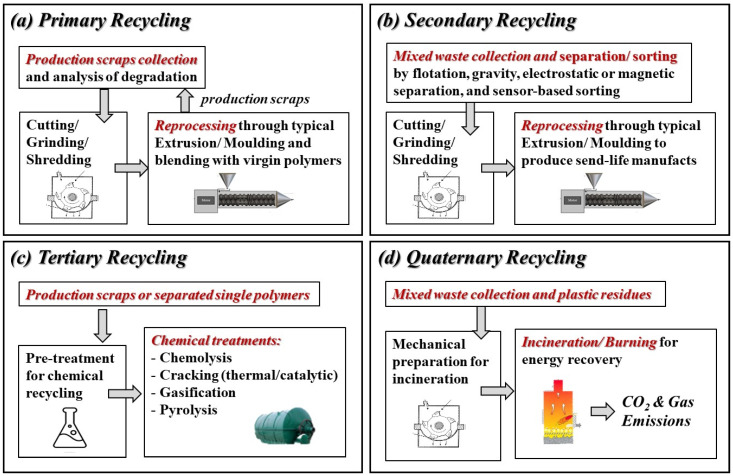
Overall illustrations of current strategies for the recycling of (bio)polymers and their composites.

**Table 1 polymers-14-02038-t001:** Overview of recycling methods for thermoplastic materials.

Recycling Methods	Short Description	Advantageous (+)/ Disadvantageous (−)	Status of the Technology	References
***Primary recycling,*** i.e., mechanical recycling of production scraps	***closed-loop recycling***: recovered polymer materials must be cut/ crashed/ grinded/ shredded to small-sized pieces and reprocessed with virgin materials	(+) post-consumer and virgin polymers have same chemical nature (+) post-consumer polymers are separated from contaminants (−) post-consumer polymers must be stabilized against degradation during reprocessing	industrially applicable practice	[11,12]
***Secondary recycling,*** i.e., mechanical recycling of waste	***polymers separation techniques and reprocessing***: (i) separation techniques by floatation, gravity, electrostatic or magnetic separation, and sensor-based sorting; (ii) cutting/crashing/grinding/shredding to small-sized pieces; (iii) reprocessing by traditional processing techniques, such as extrusion, injection/compression molding; etc.	(+) separations based on different principles: floatation, gravity, electrostatic or magnetic separation, and sensor-based sorting (−) polymers experience different degradation/oxidation levels in service (−) mixed waste contain contaminants and they are incompatible (−) second-life materials show low properties	industrially applicable practices, that continuously evolved, considering the changes of waste stream compositions	[13,14,15,16,17,18,19,20,21,22,23,24,25,26,27,28,29,30,31]
***Tertiary recycling***, i.e., chemical recycling through chemolysis, cracking, and gasification	treatments for bonding scission through chemicals, heat with and without catalytic agents, for mono-/oligomers recovery	(+) second-life materials show excellent properties (−) high-cost technology (−) negative environmental impacts due to use of chemicals, solvents, etc.	under investigations; industrially applicable for PET	[32,33,34,35,36,37,38,39,40,41,42,43,44,45,46,47,48,49,50,51,52,53,54,55,56,57,58,59,60,61,62,63,64,65,66,67,68,69,70,71,72]
***Quaternary recycling***, i.e., energy recovery	incineration of mixed plastic to recover their embedded energy	(+) energy recovery, which is preferable to landfilling and disposal in the seas and oceans (−) gas emissions must be controlled to minimize their negative environmental impacts	applicable	[73,74]

**Table 2 polymers-14-02038-t002:** Overview of recycling methods for thermoset polymers.

Recycling Methods	Short Description	Advantageous (+)/ Disadvantageous (−)	Status of the Technology	References
***Tertiary recycling,*** i.e., chemical recycling through chemicals and/or energy	***chemical recycling*** consisting of destroying the three-dimensional crosslinked structure	(+) materials recovery (−) negative environmental impacts due to use of chemicals, solvents, etc.	under investigations; limited industrially applicable practice	[75,76,77,78,79,80,81,82,83,84,85,86,87,88,89]
***Quaternary recycling***, i.e., energy recovery	incineration of three-dimensional crosslinked plastic to recovery of their embedded energy	(+) energy recovery, which is preferable to landfilling (−) gas emissions must be controlled to minimize their negative environmental impacts	applicable	[75]

**Table 3 polymers-14-02038-t003:** Overview of recycling methods for thermoplastic and thermoset micro-/nano-composites.

Recycling Methods	Short Description	Advantageous (+)/ Disadvantageous (−)	Status of the Technology	References
***Primary and Secondary recycling,*** i.e., mechanical recycling	***closed-loop recycling and downcycling*** recovered composites must be cut/crashed/grinded/shredded to small-sized pieces and reprocessed	(+) materials recovery (−) second-life materials show low properties (−) high-cost technology (in same cases it needs different processing technology)	under investigations; limited industrially applicable practice	[105,106,107,108,109,110,111,112,113]
***Tertiary recycling,*** i.e., chemical recycling through chemicals	***chemical recycling*** consisting in separation of matrix and reinforced particles/fibers	(+) materials recovery (−) negative environmental impacts due to using of chemicals, solvents, etc.	under investigations;	[114,115]
***Quaternary recycling***, i.e., energy recovery	incineration of composite materials	(+) energy recovery (−) gas emissions must be controlled to minimize their negative environmental impacts	under investigations; applicable	[116,117,118,119]

**Table 4 polymers-14-02038-t004:** Overview of recycling methods for bioplastics and bioplastic-based micro-/nano-composites.

Recycling Methods	Short Description	Advantageous (+)/ Disadvantageous (−)	Status of the Technology	References
** *Primary and secondary recycling* **	reprocessing	(+) materials recovery (−) second-life materials show low properties (−) high-cost technology	under investigations;	[122,123,124,125]
** *Tertiary recycling* **	using chemicals and/or micro-organisms	(+) materials recovery (−) negative environmental impacts due to use of chemicals, solvents, etc.	under investigations;	[126,127]

## Data Availability

The data presented in this study are available on request from the corresponding author.

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
