# Peer review of "Recycled (Bio)Plastics and (Bio)Plastic Composites: A Trade Opportunity in a Green Future"

_polymers, 2022, doi:10.3390/polym14102038_

Round 1
Reviewer 1 Report
This review summarized the important field of plastic recycling. This is a major hot topic in recent years in the field of green chemistry and polymer chemistry. The authors adequately demonstrated the importance of the topic and the review is well-written and informative. Therefore the reviewer recommends the publication of this review in Polymers with the following comments:
- The following review offers in depth summary of catalytic chemical recycling of commercial plastics: Mater. Horiz., 2021,8, 1084-1129. The authors might find this review useful for the introduction section of this work.
- For PET catalytic deconstruction, advance work has shown effective hydrogenolytic deconstruction of PET in high yields: for example, Marks et al. doi.org/10.1002/anie.202007423. The authors might make reference to these works and add discussions to this review in section 2.2, for example.
- Figure 1 and 2 are great examples of using figures/schemes to guide a clearer review article. Perhaps the authors might want to consider using more figures/schemes/tables to help summarize the rest of the review. The use of useful, attractive figures/schemes will definitely help future readers to understand the concepts.
Author Response
Thank you of reviewer for his/her positive evaluation and recommendations. All suggestions have been addressed and all corrections that we made in the text, are reported in revision mode.
- Recommended papers have been cited and commented in revised manuscript.
- New Tables 1-4 and Figures 1&4, that facilitate and guide the readers to a better understanding of review, have been included in revised version.
Please, see revised text (in revision mode) and new Tables 1-4 and Figures 1&4.
Thank you again for suggestions, the quality of our work has been significantly improved.
Reviewer 2 Report
The manuscript "Recycled (bio)plastics and (bio)plastic composites: a trade opportunity in a green future" shows main process how nowadays plastics are separated and some being recycelt.
Also the title doesn't match so much the content as most focus orientate on how plastics in nowadays industry are separated and which procedure is applied. So far there are many and the readers get lost in such without some figures given as example.
The main issue is the readability as it is pure text and it need to be somehow made more attractive for readers.
Please add besides the technical terms in the introduction (Table would be great) how much plastic is recycled (in general in industrial about thermo plastics at 1%).
Where are the problems, how can it be solved, what advantages have biodegradable etc. Those are attract the readers to read further
There only one figure 1 given in the manuscript which explains the basics how waste is separated.
- please include figures where some process can be shown for the major recycling techniques
- Please include tables that can give some comparison as the most widely used techniques and their efficiency. The issue of PSS also not addressed in landfill and their toxic nature. Additionally the time such polymers degrade in nature (several thousand years) need to be in some form mentioned.
- Reduce the method of separation in waste and give more details and description for natural polymers as example starch, cellulose and rubber is missing and other applied already in industry.
- the micro and nano plastics comes a bit short as well as it became major concern especially to remove those from waste water.
- The toxicology of such waste plastic in nature for flora and fauna as well humans are as well another part that highlights the needs to get rid of synthetic polymers.
- An outlook which developments most promising would be beneficial
Author Response
Thank you of reviewer for his/her positive evaluation and recommendations. We follow all suggestions and all corrections that we made in the text, are highlighted in revision mode.
1-2. New Tables 1-4 and Figures 1&4, that facilitate and guide the readers to a better understanding of review, have been included in revised version. New Figure 1 reports useful data regarding the polymer waste treatments at EU level, specifically, how of plastics waste (in %) are subjected to incineration, recycling and landfilling, while, new Figure 4 illustrates the current recycling strategies.
- The authors think that the separation methods for all kind of polymer-based materials, being a crucial point for the quality of recovered materials and/or second-life manufacts, needs to be addressed and discussed widely. For this reason, we prefer to maintain initial large description. Therefore, revised manuscript contains numerous changes, and we hope that in this version, the manuscript could be considered attractive for readers and suitable for publication on Polymers.
- The problem of recycling for micro-/nano- composites will become more important in the next decade, considering current large use of these materials. Unfortunately, these materials are not considered for recycling at large industrial scale.
- The harmful effect of micro and nano plastics has been shortly commented in the introduction section of revised manuscript.
- An additional comment on the statistical data for EU countries, regarding recycling, landfilling and incineration (see new Figure 1), and a comment on the future of materials recycling (see revised conclusion section; last part) have been added in revised manuscript.
Please, see revised text and new Tables 1-4 and Figures 1&4.
Thank you again for suggestions, the quality of our work has been significantly improved.
Round 2
Reviewer 2 Report
The authors made changes and the manuscript is now in much better form. the reviewer suggest accept as it is.